# Insights into the Edible and Biodegradable Ulvan-Based Films and Coatings for Food Packaging

**DOI:** 10.3390/foods12081622

**Published:** 2023-04-12

**Authors:** Huatian Wang, Zhen Cao, Lingyun Yao, Tao Feng, Shiqing Song, Min Sun

**Affiliations:** School of Perfume and Aroma Technology, Shanghai Institute of Technology, Shanghai 201418, China

**Keywords:** ulvan, green algae, edible films and coatings, food packaging

## Abstract

Recently, edible films or coatings that are made from algal polysaccharides have become promising candidates for replacing plastic-based packaging materials for food storage due to their non-toxic, biodegradable, biocompatible, and bioactive characteristics. Ulvan, a significant biopolymer with unique functional properties derived from marine green algae, has been extensively used in various sectors. However, there are fewer commercial applications of this sugar in the food packaging industry compared to many other algae-derived polysaccharides, such as alginates, carrageenan, and agar. This article aims to review the unparalleled chemical composition/structure and physiochemical properties of ulvan and the latest developments in ulvan-based edible films and coatings, thus highlighting their potential applications in the food packaging industry.

## 1. Introduction

Recently, there has been increased interest in exploiting biopolymers that are made from an array of agricultural commodities and/or food waste products due to concerns about limited fossil fuel reserves and the environmental impact caused by plastic-based packaging materials. These packaging materials are preferable because they have better material properties [1,2]. They have synthetic structures that are safe, convenient, and economical [3]. However, a major disadvantage of plastic-based packaging is that it is non-biodegradable and non-renewable [4]. Alarmingly, the production of plastic materials globally has exceeded more than 8.3 billion tons since the early 1950s, but only 9% of this waste has been recycled [5]. It has been estimated that more tons of plastic litter will end up in the oceans by 2025 [6], which will create a phenomenon called ‘plastic soup’. Only less than 1% of plastic floats on the surface of the sea, whereas most plastic floats deeper in the water column or sinks to the sea floor, posing a big threat for the environment and human health [7]. Thus, researchers are now focusing more on environmentally friendly alternatives for the packing industry.

Biodegradable packaging materials are divided into biodegradable films and coatings, and are considered an alternative to replace plastic packaging materials because they can be consumed with the product or can be decomposed by microorganisms without generating hazardous environmental emissions [8]. The production of biodegradable packaging for food packaging applications constitutes a market segment of high interest due to environmental and societal concerns in recent years [9]. Primary materials that have been used in the production of biodegradable packaging materials are polysaccharides, proteins, and lipids [10]. These biopolymers are mainly from edible sources, such as corn, wheat, cassava, potato, and sweet potato starch. However, the use of theses sources may increase food insecurity concerns among consumers since most food items come to market in food packaging [11]. Therefore, it is of significance to consider other sources of biodegradable materials beyond these agricultural products. Biopolymers from marine sources stand out as a promising matrix for producing packaging for food preservation.

Marine macroalgae or seaweed are used for several industrial applications and for human consumption [12]. Some natural polysaccharides from seaweed, including agarose, ulvan, and fucoidan, are widely utilized in the fields of food technology, biotechnology, microbiology, and medicine, and have gained a growing interest in the packing industry as a sustainable and renewable bio-resource [13,14]. There are many relevant studies exploring this; however, the sulfated polysaccharides they focused on are mainly brown and red algae, namely fucoidan, carrageenan, and agar [15,16,17,18,19]. Ulvan is from green seaweed (species from genus *Enteromorpha*) and has attractive gelling properties [20,21], great antioxidant activity, good barrier and optical properties [22,23], and still remains relatively underexploited in the packing industry, especially for food packaging. Therefore, the purpose of this review was to provide readers with an insight into the distinctiveness of this sulfated polysaccharide in terms of its physiochemical characteristics and various bioactivities, and to outline the latest developments and the potential of biodegradable ulvan-based films and coatings for food packaging.

## 2. Ulvan

### 2.1. Structure and Physiochemical Properties

Ulvan refers to water-soluble sulfated polysaccharides from members of the *Ulvales*, and are mainly observed in *UlVa*, *Enteromorpha* sp. [24]. The ulvan polysaccharides are composed of sulphate groups and unusual sugar residues, with a high content of ionic groups, high water solubility, and unique rheological properties [25], which displays several interesting structural and functional properties and are markedly different from the polysaccharides found in higher plants.

Ulvan polysaccharides are essentially composed of sulfate, rhamnose, xylose, and glucuronic acid, as reported in the early 1950s to 1960s [26,27]. However, its monosaccharide composition was further reported in the 1990s, which was found to be rich in rhamnose (16–29%), glucuronic acid (8–16%), and iduronic acid up to 8–9% [28]. These rare sugar residues contained in ulvan polysaccharides stand for strikingly different characteristics from the residues observed in other algal polysaccharides. Therefore, numerous work has been devoted to the structure and monosaccharide composition of ulvan ever since then, as the presence of iduronic acid and sulphated rhamnose within the backbone of ulvan is a distinguishable feature. These rare sugars (residues) in the ulvan polysaccharidic chain stands for a strikingly different characteristic that has yet to be identified in other algal polysaccharides. Aldobiuronic acids β-D-GlcpA-(1→4)-L-Rhap 3-sulphate and α-L-IdopA-(1→4)-α-L-Rhap 3-sulphate are the main disaccharide units repeating in the structure of *Ulva* ulvan (shown in the Figure 1) [15]. In 2007, Lahaye and Robic reviewed the monosaccharide composition and structural characteristics of ulvan polysaccharides, and suggested that the composition and structure of ulvan were mainly dependent on the algal species and origin [24].

Meanwhile, ulvan is known to have great metal ion binding ability, low viscosities, a gel-forming ability in aqueous solution [24], and is reported to have different molecular weights ranging from 1.5 × 10^5^ to 2 × 10^6^ Da. The main functional and biological properties of ulvan are presented in Figure 1. In recent work, the great metal ion binding property of ulvan has been investigated, which might be attributable to the presence of functional groups (i.e., hydroxyl and carboxyl groups) contained in ulvan that may chemically interact with metal ions during the adsorption process [29]. For its gel-forming ability, ulvan can form gels in aqueous metal ion solutions (such as 0.1 M NaCl), owing to the presence of ionic species and regular arrangements of aldobiuronic sequences that exist in the polysaccharide, which would be novel and promising packaging materials for the pharmaceutical and food fields [30]. In addition, other metal ions, such as Cr^2+^, have also been shown to have interactions with ulvan polysaccharides and would result in metal-ion-induced gelation due to the crosslinking of ulvan chains [29]. Furthermore, varying extraction processes, such as chemical extraction, enzymatic extraction, or the combination of the two methods, and different parameters, such as pH and temperature, have been reported to greatly affect the above-mentioned properties of ulvan during extraction procedures [31,32].

### 2.2. Biological Activities

It has been well established that ulvans possess a broad range of excellent bioactivities, including antioxidant [33], anti-viral [34], anti-tumor [35], anti-coagulant [36], anti-hyperlipidemic [37], and immuno-stimulating activities[38]. Nevertheless, the bioactivities of ulvan polysaccharides are mostly affected by their sugar composition, functional groups, molecular weight [33], sulfate content, and conformation [39,40]. Fernández-Díaz [41] found that the immune response in fish macrophages could be enhanced by delivering ulvan-loaded nanoparticles, which indicates the fact that ulvan can be considered an active ingredient in activating macrophages due to its good immune-stimulant activity. Adrien [36] compared the anticoagulant activity of the sulfated ulvan with some commercial anticoagulants (heparin and Lovenox (R)). The results demonstrated that the anticoagulant activity of ulvan fractions are higher than that of Lovenox (R) against both the intrinsic and extrinsic coagulation pathways and could be used as a therapeutic agent. In the meantime, ulvan is able to recover abnormal levels of oxidative stress and cytokines, and helps to boost the anti-inflammation, as well as anti-oxidation. Li [42] revealed that ulvan has an alleviative effect on inflammatory bowel disease (IBD) by reducing the DSS-induced disease activity index, colon shortening, and colonic tissue damage. We still need to conduct some targeted bioactive studies using modern technology in order to explore and promote the potential industrial applications of ulvan.

### 2.3. Industrial Applications

Despite the fact that green algae has been consumed as food for centuries, it did not gain industrial interest in other areas until the discovery of ulvan, a prominently bioactive constituent of this species. Ulvan polysaccharides have been majorly exploited in the food, pharmaceutical, and biomedical industries [43] in the form of membranes, particles, hydrogels, 3D porous structures, or nanofibers, etc. (Figure 2). From a food consumption perspective, they can be used as prebiotics in synbiotic yogurt production, which can significantly increase the natural flavor and stimulate the growth and activity of probiotic bacteria [44]. Morelli [45] developed ulvan-based emulsions, which can be used for body cream milk products as stabilizing and perfuming agents. They can also be used in soft drink as clouding and flavoring agents. With respect to biomedicine, biocompatible, non-toxic, and biodegradable ulvan can be extensively used in biomedical devices [46]. For instance, Dash [47] developed a photo-cross-linked polymeric scaffold based on ulvan through mineralization for bone tissue engineering. The presence of anionic groups on ulvan’s backbone play a significant role in affecting mineralization. Alves [48] designed a 3D porous architect based on cross-linked ulvan, which demonstrates the adequate mechanical integrity of ulvan-based biomedical materials that can withstand outer forces and maintain the physical integrity of tissue. In addition, ulvan with chemical modifications, such as carboxymethylated treatments, could produce improved medical materials. According to Barros [49], the mechanical performance of the polymeric component of bone cement formulations based on carboxymethylated ulvan was enhanced. Don [50] also demonstrated a kind of micro needle derived from ulvan that featured rapid dissolution and penetration into skin. This evidence suggested that ulvan could constitute part of a potential and efficient drug delivery system for use in the pharmaceutical industry. Overall, the current industrial applications of ulvan are still limited, despite its broad range of bioactivities and physiochemical properties.

## 3. Properties of Ulvan-Based Films and Composites

Ulvan from green seaweed can be used in the formation of films that are able to obstruct the evaporation of water, oxygen, and ultraviolet light to some extent, and render great optical properties and certain mechanical properties, such as tensile strength and elongation at break [22]. Factors that influence the physiochemical, optical, barrier, and mechanical properties when designing ulvan-based films and composites include the variation of additive types (crosslinking agents, plasticizers, antioxidant agents, etc.) and concentration [22], the chemical properties of the polysaccharide (molecular weight, charge distribution, conformation, hydrophilic behavior, etc.), and the film processing conditions (temperature, pH, and solvent used, etc.).

Reportedly, ulvans are unable to form membranes independently with ideal mechanical properties, barrier properties, and thermal stability. It needs to be improved or intensified in order to be equivalent to other petroleum-based synthetic materials with strong mechanical strength. Thus, current research has focused more on ulvan composite films, which are incorporated by additives, such as crosslinking agents and plasticizers, in an attempt to ameliorate the functional properties of the films while maintaining their overall biological activity. Don [51] prepared complex films based on the crosslinking of chitosan and ulvan. The swelling ability and water vapor transmission rate of the films were enhanced compared to pure chitosan or ulvan films, whereas consistent bioactivities, such as antioxidant and whitening activities, were also observed. The tensile strength of the complex film was increased by adding tripolyphosphate and glycerol. Toskas [52] discovered that the combination of two polymers, such as ulvan and chitosan, leads to the formation of stabilized membranes because of electrostatic interactions in acidic conditions. Therefore, it might be concluded that composite films that are made up of biopolymers, such as ulvan and chitosan, can be effective alternatives for synthetic packaging materials. Furthermore, the approach of incorporating one or more active ingredients in such composite films can enable them with better properties. As Don demonstrated, the addition of certain amounts of tripolyphosphate as an ionic crosslinking agent and glycerol as a plasticizer can prominently improve the tensile strength of the chitosan/ulvan complex films. In the meantime, he found that the addition of chlorophyll to films showed a synergistic effect with ulvan in enhancing the antioxidant activities of film. Notably, the amount of additives incorporated into the film need to be evaluated carefully in order to suitably enhance the complex film properties, while reducing the cost of preparation. On top of that, changes to the physical and rheological properties of the resulting coatings and films may occur due to compatibility or incompatibility between the incorporated biopolymers [53]. Guidara [54] discussed the effect of the type of plasticizer on the properties of the resulting ulvan films. For example, a plasticizer such as glycerol could decrease the moisture content of films compared to those supplemented with sorbitol, despite its increased mechanical properties. This phenomenon results from reduced interactions of hydrogen bonds between ulvan segments and glycerol, thereby decreasing the hydrophilic character and enhancing the moisture resistance of the composite films. Glycerol, as Blanco-Pascual [55] has discovered, can efficiently link to uronic units. Thus, it can formulate a more compact structure together with ulvan, leading to a film with a stronger barrier to water vapor. This might lead to the assumption that the water resistant values of polysaccharide-based films are significantly influenced by the residual presence of hydrophilic polysaccharides and uncross-linked fractions.

Up to the present day, despite numerous studies on the development of other seaweed polysaccharide-based films and composites, reports regardging the preparation of ulvan-based films are limited, and it seems that combining ulvan with extra film-forming biopolymers will be a more prevalent choice for industrial applications.

## 4. Edible Films and Coatings Developed from Ulvan Polysaccharides for Food Packaging

### 4.1. Novel Food Packaging

Food packaging plays an important role in preventing the contamination and deterioration of food, and thus maintains its quality during long-term transportation and storage [56]. Additionally, it provides consumers with ingredient and nutritional information [57]. Synthetic polymers that originate from petroleum resources are common materials used in the packaging of food and have been used for decades; however, they are resistant to all forms of degradation and are expensive [56]. Natural polymer-based biodegradable film materials have attracted more attention in the packaging industry. Films made from biodegradable polymers are generally considered novel food packaging (NFP) systems that are different to traditional food packaging (TFP), which is just a non-functional physical barrier protecting against chemical, physical, and microbial damage [58].

Novel food packaging emerged in response to consumers’ desire for convenient, ready to eat, tasty, and minimally processed food products [59]. NFP, featuring rough information about the quality and nutrients of packaged foods and that control microbial growth, moisture, and oxidation, are divided into two new categories: active and smart packaging systems [60,61]. This new-generation packaging is more helpful in providing consumers with direct information regarding food quality compared with TFP through certain indicators in the packaging system that monitor the quality of the food (color, smell, etc.) and safety. For instance, active packaging systems are able to work to scavenge oxygen from the package or the product and may be activated by an outside source, such as UV light [62]. Novel food packaging systems include biomaterials [63], nanomaterials [64], microbiology [65], nanoemulsion, and encapsulation [66]. These technologies play a vital part in innovative food packaging systems. Edible films and coatings are one of the aforementioned novel packaging systems that use technology that we will discuss later in this review. The main functions of these edible packaging systems are illustrated in Figure 3.

### 4.2. Edible Polysaccharide-Based Films and Coatings

Edible films and coatings are an important subject in innovative packaging systems that are functioning to delay the movement of water, gas, solvents, and oils in order to enhance structural stability, to mask volatile flavor compounds, and to carry food additives [67]. There are many products, including cereals, spices, meats, drinking water, chocolate and confectionery, fruits and vegetables, marine products, and many more, that can be effectively and economically packed by edible films and coatings [68]. Edible films are obtained as thin layers that are used to wrap food products, whereas edible coatings are applied in the liquid form of the film-forming dispersion [69]. The greatest benefits of these edible films and coatings are their biodegradability and their edibility [70] in contrast to synthetic packaging films. The problem facing synthetic packaging films in particular is about environmental issues in the fields of recycling and incineration [71]. Since many characteristics of edible films and coatings are similar to those of their synthetic counterparts, they can partially or totally supplant conventional packaging materials [72]. The principal components that are used to produce edible/biodegradable films or coatings include proteins, polysaccharides, lipids, possible combinations of these, solubilizing medium (water, ethanol, etc.), and plasticizers, etc. [73,74].

Edible films and coatings are prepared from solutions that constitute different component groups: a matrix that is formed by a cohesive structured biopolymer obtained from natural sources; additives like plasticizers, cross-linkers, surfactants, and nano-reinforcements that enhance the functional properties of the packaging; and a solvent (water or ethanol) [75]. Methods that are traditionally used for processing edible films can be divided in two main groups: wet and dry processes [76]. Dry methods, including extrusion, injection, blow-molding, and heat-pressing processes [77], are most commonly used for industrial scale-up, whereas the wet process that need solvents for the formation of the edible film are mainly utilized in the lab. Fabra [78] managed to develop a novel active edible alginate film through the solvent casting method, where lipids were added into films in order to improve their water vapor permeability. Similar methods have also been reported by Seslija [79].

Polysaccharides, a class of natural macromolecules, have the tendency to be extremely bioactive, and have mainly been used for food coating due to their excellent selective permeability to oxygen and carbon dioxide [80]. They also present remarkable mechanical and optical properties; however, they are highly sensitive to moisture, resulting in poor water vapor barrier properties [71]. Theses eventual polysaccharide-based films, due to the hydrophilic properties of the polymers, have poor moisture barriers, selective permeability to O_2_ and CO_2_, and resistance to oils [81]. However, incorporating components, such as cross-linking agents, colorants, surface active agents, lipid-based materials, or antimicrobial agents, into the film matrices can otherwise significantly improve the properties of these films through interactions with each other [74]. The functions of the major components added to the film are listed in the Table 1. Yuan [82] constructed a novel edible composite film based on chitosan with increased thermal stability and tensile strength after incorporating W_1_/O/W_2_ double emulsions as active agents. Coating salmon fillets with this edible film significantly increased the shelf life of these fish fillets, demonstrating persistent antibacterial activity during the preservation period. Navarro [83] encapsulated thyme oil, as an antimicrobial compound, into an active edible coating in order to improve the shelf life of coated products. The results showed that the selection of an encapsulating agent could affect the antimicrobial properties, rheological behavior, particle size, and stability of the film-forming emulsion, indicating a possible interaction between the encapsulating agent and the antimicrobial components. Therefore, optimization of the functional properties of edible films by means of combining the advantageous properties of each compound and the synergy effect could potentially become a main strategy to make up for the weaknesses of polysaccharide-based edible films. Furthermore, considering the properties of specific products to which films are applied, their compatibility [84,85] should be taken into account in the formulation process.

### 4.3. Preparation of Ulvan-Based Edible Films or Coatings

Marine-derived polysaccharides, such as alginates, carrageenan, and agar, can impart the hardness, crispness, compactness, viscosity, adhesiveness, and gel-forming ability of the films [100] and can enhance the sensory attributes of food products. Therefore, they are commonly selected as the perfect substrates to form the edible films and coatings. Recently, Ulvans derived from green seaweed have increasingly been studied in order to formulate biodegradable active edible films or coatings due to their various excellent bioactivities and their biocompatibility, which is akin to the synthetic polymers. For the time being, reports concerning ulvan-based edible films or coatings are predominantly focused on the preparation of the films and the active effects they have on the organoleptic characteristics of food. There is less emphasis on the mechanisms of the physical or chemical interactions between food and ulvan-based packaging materials.

Currently, the literature on the preparation of ulvan-based edible films and coatings, depending on their specific applications, is still limited. The available papers suggest that edible films, with the combination of various biopolymers and film enhancers, have better mechanical and barrier properties compared with ulvan-only-based packaging. In particular, the edible films formed by blending different functional algal polysaccharides [101] has become a focal point. This kind of composite edible film offers mechanical and chemical resistance to oxygen [102], moisture, and oils compared to the sole film, when serving as a carrier of active substances containing antioxidants, flavors, colors, and anti-bacterial agents, and can also improve the mechanical integrity of whole food products [103]. Ulvan, with excellent bioactivities and biocompatibility, has always been an essential part in making smart or active edible packaging films. Ganesan [23] designed a composite edible film based on carrageenan and ulvan polysaccharides with ulvan as an added active antioxidant agent and a major film component, whereas semi-refined carrageenan, which contained the partial derivatization of protein alongside hydrocolloid, enhanced the gelling properties of the film solution. This innovative combination not only noticeably improved the physicochemical and mechanical properties of the film, but was also a great vehicle for the better value addition of ulvan as natural antioxidant, which can effectively prevent oil oxidation and the formation of rancid flavors in lipid-based food products. Similarly, Gomaa [104] utilized cellulose and ulvan to produce composite edible films with promising natural antioxidant properties, good barrier properties, and thermal stability through hydrogen bonding interactions between cellulose and ulvan. Guidara [22] indicated the role of film-forming additives and extraction procedures for the stability and characteristics of ulvan film, and ascertained that ulvan was successful for film formulation regardless of the extraction procedure, whereas the optical, thermal, structural, and antioxidant characteristics of the film were affected by the type and concentration of additives. Additionally, some unwanted characteristics of the food, such as the loss of weight and constituents [105], because of the potential interactions between food and its packaging (transfer of chemical substances across the food–package interface, corrosive action by a metal package, or microbiological contact with packaging materials), [106] could happen during food storage and need to be carefully monitored.

## 5. Discussion

In light of the current severe environmental problems brought about by plastic-based food packaging [1], intensive and constant efforts are being made by several researchers to seek new packaging alternatives [107], which would likely be used for maintaining overall sensorial quality, enhancing the shelf life of fresh food commodities, and protection against foodborne pathogens that deteriorate food quality. Ulvan, which possesses great biodegradability and remarkable biological activities in combination with its tunable physicochemical and rheological properties [108], has the significant potential to replace synthetic macromolecules in the packaging industry. Nevertheless, investigations into ulvan-based biodegradable packaging for meat, seafood, etc., are still limited. The studies presented here have demonstrated a number of ulvan’s unique physiochemical properties, including certain gel-forming abilities and excellent bioactivities, such as antimicrobial and antioxidant properties, which could make it suitable for the formation of food wrappings and films. Additionally, some of the properties and abilities of ulvan-based films and coatings can be modified and controlled through the incorporation of other natural polymers, leading to better mechanical properties and barrier characteristics and making it more convenient for food storage. In all, the application of ulvan as a renewable alternative in the food packaging industry is expected to rapidly increase in the future. Nonetheless, more extensive research into the safety of ulvan for human consumption and the effects of potential interactions between ulvan-based edible films and foodstuffs on the sensorial properties of commercial products are required for the purposes of safe consumption.

## 6. Conclusions

Ulvan, with interesting and unique properties, has increasingly been gaining attraction in various industries, such as the food, medicine, and cosmetic industries, etc. Against the backdrop of severe plastic pollution across the world in recent years, ulvan has the potential to make edible films and coatings that replace synthetic plastic packages thanks to its biodegradability, non-toxicity, and excellent bioactivity. However, current research about formulating ulvan-based films in this field are rather limited. Thus, in this paper, we have explored the significance of ulvan-based edible films as to their satisfactory performance in food preservation, with the aim of increasing the possibility of their use for food packaging. Currently, for the purpose of further expanding ulvans’ use in the food packing industry, we still require adequate studies to elucidate the mechanisms of ulvan-based film formation and possible interactions between the film and food components during preservation.

## Figures and Tables

**Figure 1 foods-12-01622-f001:**
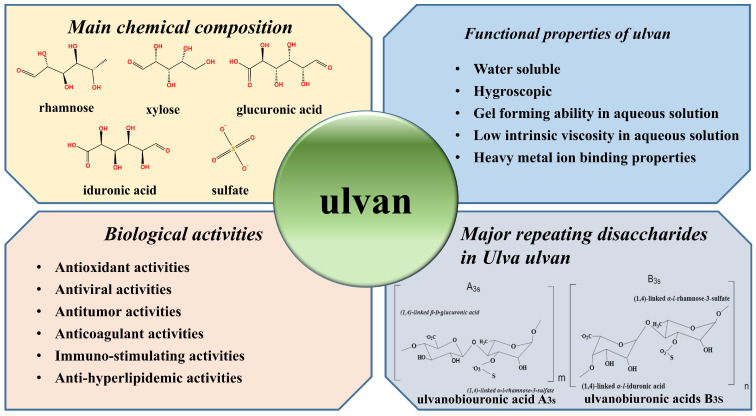
Structural and physiochemical properties.

**Figure 2 foods-12-01622-f002:**
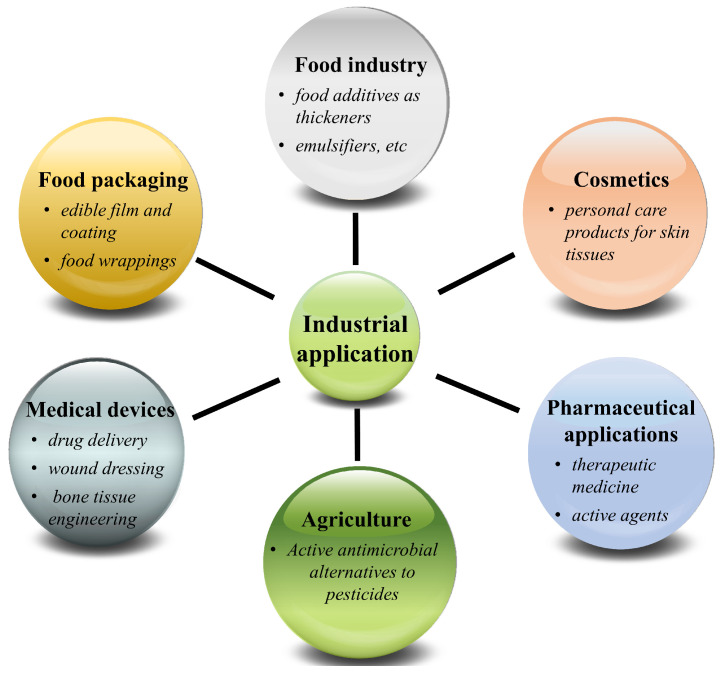
Major applications of ulvan polysaccharides in various industrial fields.

**Figure 3 foods-12-01622-f003:**
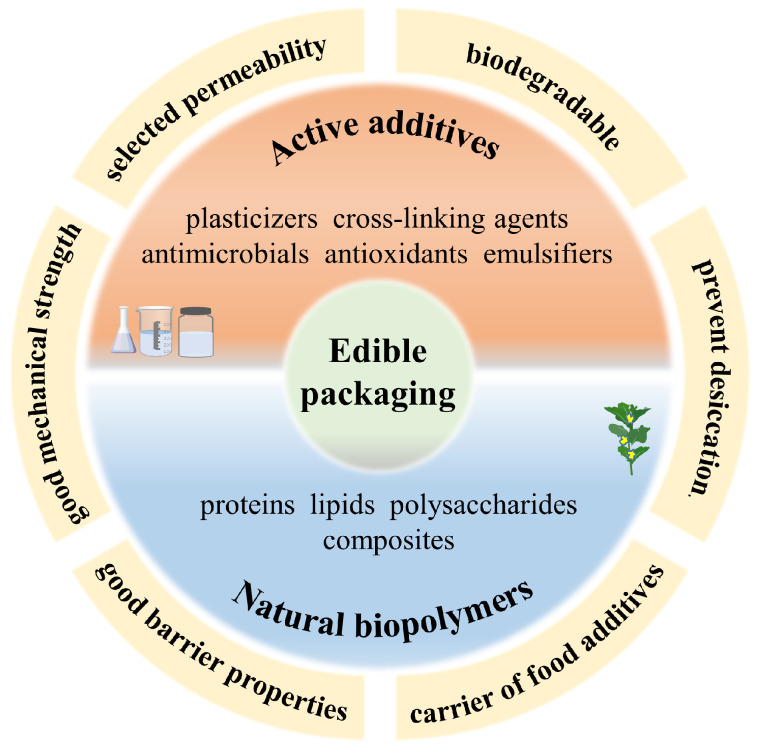
Major structural matrix and characteristics of edible films and coatings.

**Table 1 foods-12-01622-t001:** Role of common additives added to some polysaccharide-based edible films and coatings.

Film Polymers	Major Additives	Function	Reference
K-carrageenan, i-carrageenan, and alginate	Glycerol	Plasticizer Film enhancer	[86]
Ulvan	Glycerol	Plasticizer Film enhancer	[87]
	Polyethylene glycol (PEG)	Plasticizer Film enhancer	
Sodium alginate and agar	Essential oils	Antimicrobial agent	[88]
		Antioxidant agent	
Phycocolloid (carrageenans/porphyrans)	Glycerol	Plasticizer Film enhancer	[89]
	Ca^+2^	Stabilizer	
Gracilaria chouae polysaccharide	Carboxymethylcellulose (CMC)	Film-Forming Additive;	[90]
	Lysozyme	Antibacterial agent	
K-carrageenan and alginate	Cellulose nanofibers (CNF)	Film-Forming Additive;	[91]
Chitosan	Anthocyanin	pH-colorimetric indicator	[92]
Cellulose	Chitosan	Antibacterial agent	[93]
Pectin and chitosan	Tea polyphenols	Antioxidant agent	[94]
		Film enhancer	
Hyacinth bean starch	TiO_2_ nanoparticles	Reinforcement agent	[95]
Soybean polysaccharide and carrageenan	Metal nanoparticles (AgNPs)	Stabilizing agent	[96]
		Reducing agent	
Pectin and alginate	Whey Protein	Film enhancer	[97]
Gelatin	Chitosan nanoparticles	Antimicrobial agent	[98]
Soybean polysaccharide	ZnO nanoparticles	Reinforcement agent	[99]
		Antimicrobial agent	

## Data Availability

All data used within this study are included in this article.

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
