# Peer review of "Insights into the Edible and Biodegradable Ulvan-Based Films and Coatings for Food Packaging"

_foods, 2023, doi:10.3390/foods12081622_

Round 1
Reviewer 1 Report
The manuscript titled “Insights into the edible and biodegradable ulvan-based films and coatings for food packaging” is aimed to provide readers with an insight into the distinctiveness of sulfate polysaccharide in terms of its physiochemical characteristics and various bioactivities, and to outline the latest development and potential of biodegradable films and coatings based on ulvan for food packaging. The topic is relevant and interesting. However, presented manuscript should be revised for further consideration. The main comments and recommendations are listed below.
L.20-26. The authors should expand the paragraph describing undesired effects of plastic packaging. Some statistics is needed to point out the relevance of plastic packaging substitution. “Plastic soup” effect also can be discussed as the next information is linked with marine macroalgae or seaweed.
L.26. dote is missing.
Paragraph 2. Before the paragraph 2 the authors should discuss the main trends on biodegradable packaging materials. What materials are used? Novel trends and technologies? What consumers prefer? Are consumers ready for alternative packaging? Some useful sources: https://doi.org/10.1038/s41598-022-16878-w, https://doi.org/10.1038/s41598-022-10913-6. After that the authors can make the crossing to the paragraph 3 by emphasizing that materials from seaweed or microalgae biomass deserve consideration in this direction.
Figure 2. The quality of the figure is low. It should be removed or redrawn.
L. 137. …conformation,, hydrophilic behavior, etc.. (double commas)
The authors miss space before reference link basket in many cases. For example, …devices[46]…
The authors introduced some abbreviations that didn’t use further. For example, TS, TFP. This should be reconsidered.
Table 1 can be expanded. What about gelatin? chitosan? Metal based nanoparticles?
Statements in Discussion should be supported by references.
Conclusion should be expanded.
References. The topic is hot, but there are many old sources in References. The authors can change some old sources with more novel works.
Reviewer 2 Report
The purpose of the manuscript is to provide readers with an insight into the distinctiveness of the sulfated polysaccharide in terms of its physiochemical characteristics and various bioactivities, and to outline the latest development and potential of biodegradable films and coatings based on ulvan for food packaging. The topic is interesting, but the review needs some improvement:
The manuscript is missing the chemical structures of the polysaccharides present in ulvan. They are basic to explain the properties and potential applications described by the authors.
There are some paragraphs difficult to read. It seems that the authors made some improvisations, and they did not do a final check. As examples see lines 60, 62, 64, 73, 75, 78, ...., 249, 254, 311, etc. Authors must review the whole document to correct what they want to express, and after that, a style and English edition is recommended.
Table 1 is not cited in the text.
Lines 200-201. Two references (64 and 65) are cited as a source of information for the classification of novel food packaging into four categories: active, interactive, smart, and intelligent. However, the references only classify them into active and smart categories. The other categories written by the authors are just synonyms of those two.
Lines 309-317. This paragraph needs to be rewritten to express with clarity the ideas of the authors.
The conclusions section is not well written. It must focus on the findings and potential applications instead of the lacks. Additionally, formality and grammar mistakes must be corrected.
Round 2
Reviewer 1 Report
The authors considered all comments and recommendations and decided them well. The revised manuscript can be recommended for publication in Foods.
Author Response
Thanks for your comments
Reviewer 2 Report
The quality of the manuscript has improved and the requests were responded by the authors. However, there are still some misspelled words in the corrected paragraphs, as well as grammar mistakes. Since English is not my first language, MDPI English editors must correct the manuscript before the final version is published.
Additional corrections in Table 1: use capital letter in the first letter of the first word in the first column. Correct Tio2 to TiO2
Author Response
Thanks for your commments, we have made corrections in Table 1 and corrected some misspelled words as well as some grammar mistakes.